# Associations between Intake of Dietary Sugars and Diet Quality: A Systematic Review of Recent Literature

**DOI:** 10.3390/nu16111549

**Published:** 2024-05-21

**Authors:** Kelly C. Cara, Zhongqi Fan, Yu-Hsiang Chiu, Xu Jiang, Haya F. Alhmly, Mei Chung

**Affiliations:** Friedman School of Nutrition Science and Policy, Tufts University, Boston, MA 02111, USA; kelly.cara@tufts.edu (K.C.C.); zhongqi.fan@tufts.edu (Z.F.); yu-hsiang.chiu@tufts.edu (Y.-H.C.); xu.jiang@tufts.edu (X.J.); haya.alhmly@tufts.edu (H.F.A.)

**Keywords:** dietary sugars, nutritive sweeteners, diet, nutrition assessment, nutritional requirements, dietary fiber, micronutrients, vitamins, trace elements, diet quality

## Abstract

Understanding the relationship between the intake of sugars and diet quality can inform public health recommendations. This systematic review synthesized recent literature on associations between sugar intake and diet quality in generally healthy populations aged 2 years or older. We searched databases from 2010 to 2022 for studies of any design examining associations between quantified sugar intake in the daily diet and dietary indexes (DIs) or micronutrient intakes. Different sugar types and diet quality measures were analyzed separately. We converted DI results to Pearson’s *r* correlations and grouped indexes with or without a free or added sugar component to facilitate cross-study comparisons. Meta-analysis was deemed inappropriate. From 13,869 screened records, we included 27 cross-sectional studies. NUQUEST risk of bias ratings were neutral (*n* = 18 studies) or poor (*n* = 9), and strength of evidence by the GRADE approach was very low due to study design. Most studies reported negative associations for added and free sugars with diet quality indexes (*r* ranging from −0.13 to −0.42) and nutrients of public health concern (fiber, vitamin D, calcium, potassium), while associations with total sugars were mixed. Due to cross-sectional study designs, the clinical relevance of these findings is unclear. Prospective studies are needed to minimize confounding and inform causal relationships.

## 1. Introduction

Understanding the relationship between the consumption of dietary sugars and diet quality can contribute to the development of public health recommendations that can help individuals make healthier food choices and reduce diet-related chronic disease risk. Diet quality reflects dietary patterns comprised of foods and beverages that, in total, are associated with better health and reduced risk for chronic disease [1]. Many dietary indexes define high quality diets as those with higher intakes of nutrient-dense foods and lower intakes of added sugars, sodium, and saturated fat. Because our diets contain a mix of foods, research examining the impact of sugar intake on nutrient intake or diet quality is challenging, as both nutrient-rich and nutrient-poor foods are sources of sugar in the diet.

In 2015, the World Health Organization (WHO) issued a guideline on the intake of “free sugars” to reduce the risk of noncommunicable diseases for adults and children [2]. Free sugars refer to monosaccharides (such as glucose and fructose) and disaccharides (such as sucrose or table sugar) added to foods and drinks by the manufacturer, cook or consumer, as well as sugars naturally present in honey, syrups, fruit juices, and fruit juice concentrates. The WHO called for a reduction in daily free sugar intake to levels less than 10% of total energy intake. This is similar to the recommendation to limit intake of added sugars made by the committee for the 2020 Dietary Guidelines for Americans (DGA), which provide guidance to promote health and prevent chronic diseases. In 2020, the DGA committee identified added sugars as a dietary component of public health concern for overconsumption and acknowledged that the “addition of sugars to foods or beverages provides energy, generally without contributing additional nutrient intake”, but no specific quantity of sugars was recommended for chronic disease prevention. According to the DGA’s food pattern modeling, the amount of calories from added sugars that a person could consume while still meeting other dietary intake recommendations varies depending on caloric needs. After accounting for recommended intakes of other foods and food components, most Americans have less than 8% of calories remaining that could be contributed by added sugars, including added sugars inherent to healthy dietary patterns [3].

It is important to note that the impact of reduced sugar intake on health outcomes, nutrient intakes, or diet quality should be investigated using prospective study designs such as interventional or cohort studies to avoid reverse causality issues. The interrelations between intake of sugars and nutrient intakes or diet quality are difficult to disentangle in cross-sectional studies where exposures (intake of sugars) and outcomes (nutrient intakes or diet quality) are measured at the same time and often rely on self-report. Yet, most large population-based studies are cross-sectional in design.

Another systematic review previously examined associations between the intake of dietary sugars and diet quality [4]. However, the literature search end dates from that review are now more than 10 years old, and the methodological quality is considered substandard because no formal risk-of-bias assessment was performed. The overall objective of the present systematic review is to critically appraise and synthesize all available evidence on the relationship between different intake levels of sugars (added, free, or total) and diet quality in generally healthy populations. The “diet quality” outcomes of interest were pre-determined to include micronutrient intake, fiber intake, and diet quality measured by various indexes.

## 2. Materials and Methods

This systematic review followed the standards for conducting a systematic review outlined in the National Academy of Medicine’s Standards for Systematic Reviews [5], and the results were reported according to the Preferred Reporting Items for Systematic Reviews and Meta-Analyses (PRISMA) guideline [6]. A prospectively developed protocol was registered on The International Prospective Register of Systematic Reviews (PROSPERO registry: CRD42022374882).

### 2.1. Data Sources and Searches

In collaboration with a research librarian, we developed a search strategy to capture studies examining associations between quantifiable dietary sugar intake and pre-determined measures of diet quality. Search strategy terms included various names for nutritive sugars (e.g., table sugar, sucrose, honey, syrup), micronutrients (in general and specific essential vitamins and minerals), dietary fibers, and diet quality. Specific diet quality index names and acronyms were included in the search if they were not captured by general diet quality terms. Controlled vocabulary (MeSH terms in MEDLINE^®^) and standardized keywords (Emtree terms in Embase) were included in the search where applicable.

Search strategy terms were adapted for the following bibliographic databases where literature was searched from 1 January 2010, for publications in the last decade: Embase (to 27 October 2022) and the Ovid platform’s MEDLINE^®^ (to 27 October 2022), Cochrane Central Register of Controlled Trials (to 28 October 2022), and Global Health (to 28 October 2022). The MEDLINE^®^ search strategy is provided in Appendix A as an example. We also performed reference mining of studies included in relevant authoritative reports and systematic reviews. Searches were limited to English language publications, human studies, populations older than 23 months, and relevant publication types (e.g., peer-reviewed articles).

### 2.2. Study Selection

To be eligible for inclusion, studies had to quantify dietary sugars in the total diet (i.e., typical total dietary intake in a day) and report on at least one diet quality outcome of interest: micronutrient intake (e.g., essential vitamins and minerals), fiber intake, or diet quality measured by an index or as otherwise defined by the original study. Eligible study designs included interventional (parallel or crossover randomized controlled trials, non-randomized controlled trials, quasi-experimental, and before-and-after), cohort, nested case-control, case cohort, and cross-sectional. Studies conducted in generally healthy populations aged greater than 23 months (as defined by the mean or median age of the study population) were eligible for inclusion. Populations were considered generally healthy if less than 20% of participants had disease. Populations with overweights that were otherwise healthy and adults 40 years of age or older with hypertension were also considered “generally healthy”. This definition has been widely accepted by the scientific community due to the high prevalence of overweight and hypertension among older adults [7]. Table 1 presents all eligibility criteria.

Duplicate citations across databases were removed in EndNote software (Version 20.5. Philadelphia, PA, USA: Clarivate) prior to screening abstracts. Titles and abstracts were screened by two independent investigators using the Rayyan software for systematic reviews [8]. Full-text articles of screened-in abstracts were retrieved and then screened by two independent investigators. Disagreements between investigators were adjudicated by a third investigator or by consensus.

### 2.3. Data Extraction

We created standardized data extraction forms for study designs and population characteristics, with separate forms for quantitative results. Data extracted for study characteristics included design, location, duration, and funding source. Population characteristics included sample size, age (mean, median, range), sex (% male), and health status. Data were also extracted for how dietary intake was assessed throughout the study (e.g., diet assessment tool, frequency of assessment, whether and how baseline dietary data were collected and used), the study intervention or exposure (e.g., sugar type and dose, database or algorithm used to quantify sugar content of diet), and outcomes of interest (e.g., how diet quality was defined in the study, which indexes and micronutrients were reported). Finally, quantitative data were extracted for associations between sugar and diet quality outcomes. Study characteristics data were extracted by two independent investigators, and quantitative results were extracted by one investigator, then comprehensively checked by a second. Conflicts were resolved by consensus or a third investigator.

### 2.4. Risk of Bias

Though many study types were considered, only cross-sectional studies met the eligibility criteria. Currently, no tool exists to assess the risk of bias (ROB) in cross-sectional studies on nutrition topics, so we modified the Nutrition Quality Evaluation Strengthening Tools (NUQUEST) for cohort studies [9]. NUQUEST assesses potential sources of bias in the selection of study participants, comparability of study groups, ascertainment of outcomes, and nutrition-specific considerations. Our modified version excluded items irrelevant to the cross-sectional design and expanded on items related to dietary assessment tools.

Two independent investigators performed ROB assessments at the study level using ROB forms created with Qualtrics software (Qualtrics, Provo, UT, USA). Disagreements were resolved by consensus or a third investigator. The overall ROB ratings were determined based on the number of criteria that were met across domains, where a rating of “Good” meant all or almost all criteria were met (low ROB), “Neutral” meant most criteria were met (moderate ROB), and “Poor” meant that most or all criteria were not met (high ROB). Due to the inherent biases associated with the cross-sectional study design, all initial overall ROB evaluations were downgraded one level (from “Good” to “Neutral”). ROB results were considered in the interpretation and discussion of results and the strength of evidence grading.

### 2.5. Data Synthesis and Meta-Analyses

Data were synthesized separately for diet quality outcomes. Separate analyses for studies with different designs were unnecessary since only cross-sectional studies were found eligible for this review. Summary tables were created to present key study features and to facilitate qualitative synthesis. Due to the large variety of micronutrients reported by included studies, our synthesis of results focused on dietary indexes and relevant nutrients of public health concern (i.e., fiber, vitamin D, calcium, and potassium) and nutrients to limit (i.e., sodium) from the 2020–2025 Dietary Guidelines for Americans [10]. Results based on dietary indexes are presented in tables as they were reported in the original articles, but results presented in forest plots were reversed for some indexes so that all results indicate higher diet quality with higher index scores.

Quantitative data were insufficient to perform meta-analyses. To facilitate the comparisons across studies, forest plots were used to display individual study results for diet quality indexes or scores that were converted to the same scale (i.e., correlation), when possible, using an online calculator [11]. All statistical analyses and plotting were conducted with Stata/SE 18 (StataCorp. 2023. Stata Statistical Software: Release 18. College Station, TX, USA: StataCorp LLC).

### 2.6. Strength of Evidence

We planned to utilize the Grades of Recommendation, Assessment, Development, and Evaluation (GRADE) approach [12] to determine the strength of evidence (SoE) for each outcome. Since only cross-sectional studies were found eligible for this review, and cross-sectional design cannot rule out reserve causality between exposures and outcomes, SoE ratings for all outcomes were automatically deemed very low.

## 3. Results

Database searches identified 13,869 unique records, with no additional records identified through reference mining. Of these, 27 cross-sectional studies reported in 26 articles met eligibility criteria and were included in this review. One secondary analysis of an RCT reported both cross-sectional and longitudinal estimates (20-week concurrent change) [13]. The search and study selection process is outlined in Figure 1, and a complete list of excluded full-text records with exclusion reasons is reported in Appendix A.

### 3.1. Study and Participant Characteristics

Study characteristics for the 27 individual studies (from 26 articles) are provided in Table 2. Included studies assessed populations from 21 countries across all 6 populated continents, but primarily from Europe (35%) and the Americas (35%). Study funding came from government (54%), non-profit (27%), or industry sources (15%), with five reports funded by mixed sources: government and non-profit (15%) or industry and non-profit (4%). Other reports were not funded (19%) or did not mention funding (8%). These cross-sectional studies mostly utilized data from large population cohorts, and the average analyzed sample size was 5040 participants, with seven studies analyzing fewer than 1000 participants (range: 298 to 13,005). Three studies were conducted in female populations, three did not report on the sex of study populations, and the remaining 21 studies were 49% male on average. The average study population mean/median age was 25 years old (mean/median range: 2.08 to 52.56 years), with 15 studies on child/adolescent populations below a mean/median age of 18 years. One study did not report the mean or median age [14].

Most studies collected dietary data using 24-h dietary recalls (24HR, 56%), while 30% utilized food diaries and 26% used food frequency questionnaires (FFQs). Three studies used a combination of these tools [15,22,37]. Over half of the studies (59%) collected two or more days of dietary intake data (e.g., multiple 24HRs and/or diaries), while the remaining 41% collected one 24HR, a one-day diary, and/or a single FFQ. Following these assessments, studies primarily estimated the intake of dietary sugars based on national nutrient or food composition databases specific to the study country or population. The use of these databases was clearly reported by 48% of studies and presumed by another 30%. Five other studies (19%) reported estimating dietary sugars via pre-established or locally developed methods [19,22,24,33,38].

Diet quality and sugar associations (by sugar type) are summarized in Table 3 for the 26 included articles. Reported diet quality measures were based on dietary indexes or scores (42%), and dietary intake levels of fiber (58%), essential vitamins (54%), and essential minerals (62%). Types of sugars assessed in the articles were total sugars (31%), added sugars (50%), and free sugars (38%), which included naturally occurring sugars (including those from 100% fruit juices and concentrates) [24], non-milk extrinsic sugars [32], monosaccharides and disaccharides combined [34], and individual simple sugars: sucrose, fructose, glucose, lactose, and maltose [37]. Four articles assessed two or more sugar types [24,32,33,37].

### 3.2. Risk of Bias

Risk of bias criteria and ratings for each of the 27 included studies are presented in Appendix A. After downgrading initial overall ROB ratings by one level due to the cross-sectional design, 18 studies were found to be neutral (moderate ROB), and 9 studies were found to be poor (high ROB). All studies with poor overall ROB showed potential bias in the “Comparability” domain primarily because the study analyses or design failed to control for four or more factors we identified as main potential confounders: age, sex, body mass index, a socioeconomic indicator such as income or education level, and health status (if not all participants were healthy). These studies also measured dietary sugar exposure just once.

### 3.3. Synthesis of Results

The included articles assessed a variety of dietary indexes and essential vitamins and minerals as measures of diet quality in association with various types of sugars (Table 3). Results for the dietary indexes and fiber are described in detail below, along with summaries of results for other nutrients of public health concern (vitamin D, calcium, and potassium) and for sodium as a nutrient to limit. The sections below are organized by those overarching diet quality measures and by sugar type (added, free, and total sugars).

#### 3.3.1. Diet Quality Indexes

Eleven studies assessed associations between dietary sugar intake and diet quality indexes or scores [15,20,21,26,28,29,32,33,34,35,37]. Four of these studies [15,26,34,37] were rated as having high potential for bias because they failed to control for important confounders and assessed dietary intake just once (see Appendix A). Most studies included just one index or score, but some compared multiple indexes and/or included subscales. Altogether, 15 unique indexes plus 3 subscales of the diet quality measure were assessed. Eleven indexes were based on guidelines or adherence to recommended dietary intakes; two were based on dietary components predictive of chronic disease risk (Alternate Healthy Eating Index [AHEI-2010] and Global Diet Quality Score [GDQS]) [15], and two other indexes assessed dietary variety based on the number of consumed foods (Food Variety Score) or food groups (Dietary Diversity Score) [26]. Six of the unique indexes included one or more scoring components based on sugary foods or beverages, including: sugar-sweetened beverages (SSBs) [15,34], SSBs and fruit juice [15], sweets and ice cream [15], foods and beverages with easily fermentable sugars and drinks high in food acids [33], candy and snacks [34], “Other foods” mainly including those rich in added sugars or solid fats [35], and a “Sugar sources” cluster comprising fruit juice or drinks, soft drinks or energy drinks, and confectionary items [37]. Three indexes included sugar itself as a separate scoring component: all sweeteners [28], less sugar [29], and total sugars [32]. The other six indexes had scoring components based only on foods, food groups, and/or nutrients other than sugars. Appendix A presents associations between sugar intake and dietary indexes reported by individual articles. This table also presents factors accounted for in each study’s design or analysis, where most studies accounted for socio-demographic (sex, age, social class) and/or nutritional factors (energy intake, nutrient density).

##### Added Sugars

Seven studies assessed associations between added sugars and diet quality indexes [15,21,26,28,29,33,35], and two of these had poor ROB [15,26]. As depicted in Figure 2, six studies showed consistently negative associations between diet quality indexes and added sugar intake (mean Pearson’s *r* ranged from −0.13 to −0.42), while two studies showed mixed results (mean Pearson’s *r* ranged from −0.20 to 0.10), and one study showed positive associations (mean Pearson’s *r* ranged from 0.29 to 0.39). One article with poor ROB that directly compared associations with different indexes found that correlations with absolute added sugar intake (g) were consistently negative across three indexes (GDQS, AHEI-2010, and Minimum Dietary Diversity for Women [MDD-W]) regardless of whether diets were assessed by 24HR or FFQ in the Mexican National Health and Nutrition Survey (ENSANUT) and regardless of whether the indexes included or did not include a sugar component [15]. Another study conducted in adults (16–69 years) assessed adherence to various guidelines for added sugars (e.g., <5, <10, or <20%E) in association with the Dutch Healthy Diet-index (DHD-index) based on the 2006 Dutch Guidelines for a Healthy Diet [33]. When comparing males who were adherent or not adherent to sugar intake guidelines, this study found no association between added sugars and DHD-index scores. Similar results were reported for females except that DHD-index scores were significantly higher for females consuming <20%E from added sugars compared to those consuming ≥20%E (*p* < 0.001). One article with poor ROB reported a mix of positive and null associations for added sugars and three diet quality index measures in a study of children aged 4–8 years [26]. With and without controlling for energy intake, quartiles of added sugar intake (g) were positively correlated with both a Diet Diversity Score (DDS) based on the number of food groups consumed (*p* < 0.001) and a Food Variety Score (FVS) based on a total count of different foods consumed (*p* < 0.001). Without controlling for energy intake, added sugar quartiles were also positively correlated with Mean Adequacy Ratio (MAR) scores that measured recommended nutrient intakes (RNIs) for 11 micronutrients (*p* < 0.001); however, this association was null in Pearson’s partial correlations that controlled for energy intake (*p* > 0.05). In pairwise comparisons of quartiles based on %E from added sugar, diet quality scores (DDS, FVS, and MAR) showed a mix of significant positive (*p* < 0.05) and null associations. For each index in this study, diet quality scores were always lowest in the bottom sugar intake quartile, but the top two quartiles of sugar intake did not differ significantly in diet quality.

##### Free Sugars

Four studies assessed associations between free sugars and diet quality indexes [32,33,34,37], and two of these had poor ROB [34,37]. Three of these studies found consistently negative associations between intake of free sugars and dietary indexes (mean Pearson’s *r* ranged from −0.09 to −0.22), while one study found no association (mean Pearson’s *r* ranged from 0.01 to 0.06). The results are depicted in Figure 3. One study on the British Food Standard Agency (FSA) nutrient profiling system score found negative associations with non-milk extrinsic sugar as a percent of total energy (%E) in both child (4–10 years) and, adolescent subgroups (11–18 years) [32]. In Dutch children from the Generation R Study 2001–2005, a negative association was also found between intakes of mono- and disaccharides (g) and scores on an internally developed Diet Quality Score for Preschool Children (sugars: −20.28 SD increase [95% CI: 20.31, 20.26] per 1 point increase in the diet score), but ROB was poor in this study [34]. One additional study with poor ROB conducted in 15–18 year-olds from the New Zealand (NZ) Adult Nutrition Survey, 2008–2009, reported a negative association for sucrose (g/MJ) and tertiles of the Healthy Dietary Habits Score for Adolescents (HDHS-A) (*p* for trend = 0.03) [37]. Finally, one study in adults (16–69 years) assessed adherence to various guidelines for intake of free sugars (e.g., <5, <10, or <20%E) in association with the Dutch Healthy Diet-index (DHD-index) based on the 2006 Dutch Guidelines for a Healthy Diet [33]. When comparing males who were adherent or not adherent to sugar intake guidelines, this study found no association between free sugars and DHD-index scores.

##### Total Sugars

Three studies assessed associations between total sugars and diet quality indexes [20,32,37], and one of these had high potential for bias [37]. As depicted in Figure 4, one study of children and adolescents from the Australian National Children’s Nutrition and Physical Activity Survey, 2007, found a negative dose-response relationship across quintiles of total sugar intake and scores (mean Pearson’s *r* comparing Q1 to Q2, Q3, Q4, and Q5: −0.03, −0.07, −0.17, and −0.20) on the Dietary Guideline Index for Children and Adolescents (DGI-CA) [20]. One study on the British Food Standard Agency (FSA) nutrient profiling system score found negative associations with total sugars as a percent of total energy (%E) in both child (4–10 years) and adolescent subgroups (11–18 years) [32]. Another study conducted in 15- to 18-year-olds from the New Zealand (NZ) Adult Nutrition Survey, 2008–2009, found no association between total sugars (g/MJ) and tertiles of the Healthy Dietary Habits Score for Adolescents (HDHS-A), but ROB was deemed poor [37].

#### 3.3.2. Dietary Fiber

Sixteen studies from 15 articles reported on associations between dietary intakes of sugar and fiber [13,14,16,17,19,22,24,25,26,27,30,31,33,36,38], and the results are presented in Appendix A along with factors accounted for by each study’s design or analysis. Most of these studies (62.5%) were rated as moderate for potential bias (neutral ROB) due to the study design, but other studies that only had one dietary intake measure and that did not control for important confounders were rated as high for potential bias (poor ROB; see Appendix A). Most studies (75%) controlled for energy intake by including a total energy covariate in models and/or by assessing sugar as %E and fiber as %E, g/1000 kcal, or g/1000 kJ. While three other studies assessed sugar (%E), their models did not include total energy, and they analyzed fiber as a percentage of the population above the Adequate Intake (AI) level [14], as absolute intake (g) [30], or as a component score [33]. One remaining study analyzed both sugar and fiber as only absolute intakes (g) without controlling for energy [25]. Eight of the studies were conducted in child or adolescent populations [16,17,19,26,27,30,38] or reported separate results for participants under age 19 [14]. All other studies were conducted in adult populations [13,22,24,31,33,36] or combined adolescents and adults in analyses [25].

##### Added Sugars

Eight studies from seven articles reported added sugar associations with fiber intake [14,22,24,25,26,31,33], and three of these studies showed high potential for bias [25,26,31]. Five studies adjusted for energy and reported negative associations between added sugars and energy-adjusted dietary fiber intake. One article reported on two different studies of Swedish adults aged 18–80 years (Riksmaten Adults Study) and 45–68 years (Malmö Diet and Cancer Study) [22]. Both studies showed highly significant negative associations for fiber intake (%E) across six groups of added sugar intake (%E) after controlling for age, sex, BMI, and nonalcoholic energy intake. A study on Finnish adults (25–74 years) found highly significant negative associations with dietary fiber (g) across quartiles of added sucrose and fructose (g; *p* < 0.001) in both male and female subgroups while controlling for covariates (age, energy intake, leisure-time physical activity, smoking status, education, and BMI) [24]. Another study on older adults from Australia (age 49+ years) found a highly significant negative association between three levels of added sugar intake (%E) and absolute fiber (g) while controlling for age, sex, energy, and BMI, but ROB was poor in this study [31]. Finally, a study on South African children aged 4–8 years found highly significant and negative (*p* < 0.001) energy adjusted Pearson’s partial correlations for added sugars (g) and total fiber (unit unclear), but ROB was also deemed poor [26].

##### Free Sugars

Seven studies assessed free sugar associations with fiber [17,19,24,27,30,33,38], and two of these were rated as having poor ROB [17,27]. Five studies adjusted for energy, where one study conducted in adults reported positive associations, and the four other studies conducted in child and adolescent populations each reported negative associations. Finnish adults from the population-based Dietary, Lifestyle, and Genetic determinants of Obesity and Metabolic syndrome (DILGOM) study showed significantly higher dietary fiber intake in male and female subgroups, with the highest quartile of naturally occurring sucrose and fructose (g) intake compared to the lowest quartile (*p* < 0.0001) [24]. One study on Japanese participants aged 1–19 years found a highly significant negative association between four intake levels of free sugars (%E) and dietary fiber density (g/1000 kcal) after controlling for age, sex, and weight status (*p* < 0.001), but ROB was poor [17]. Another study stratified results by age group and found highly significant negative associations for quintiles of free sugars (%E) and non-starch polysaccharide fiber (g/1000 kcal) in U.K. participants aged 4–10 years and 11–18 years (*p* < 0.001) [19]. The third study in adolescents (with poor ROB) found a highly significant negative association for free sugars (%E) across quartiles of dietary fiber intake (g/1000 kcal) in Brazilians aged 10–19 years [27]. The final study included Australian participants aged 2–18 years and reported a highly significant negative association for six free sugars (%E) cutoff levels and dietary fiber (g/1000 kJ; *p* < 0.001) after controlling for covariates (age, sex, socio-economic indices of disadvantage for areas [SEIFA], equivalized household income, and remoteness of living area) [38].

##### Total Sugars

Four studies assessed total sugar associations with fiber [13,16,30,36], where one was found to have high potential for bias [16]. Three studies adjusted for energy and reported only null and/or positive associations for total sugar intake and dietary fiber. When controlling for covariates (misreporting status, sex, immigrant status, and weekend reference day), one study reported null findings for quintiles of total sugars (%E) and total fiber (g/1000 kcal) in children aged 2–8 years and 9–13 years, but a weak positive association was reported for those aged 14–18 years (*p* < 0.05) in this study with poor ROB [16]. The second study was a secondary analysis of an RCT conducted in pregnant women with a mean age of 29.8 years [13]. This study found no association between total sugars (g/1000 kJ) and absolute dietary fiber (g) after controlling for age at study entry, pre-pregnancy BMI, ethnicity, intervention group, and energy intake. The third study reported highly significant positive associations for quintiles of total sugars (%E) and total fiber density (g/1000 kcal) in adults overall when controlling for several covariates (age, sex, smoking, self-perceived health, blood pressure, diabetes, heart disease, cancer, osteoporosis, education, physical activity, income, BMI, immigrant status, weekend reference day, and misreporting status) [36]. In analyses stratified by sex and age group (19–30 y, 31–50 y, 51–70 y, and 71+ y), findings were null for males and females aged 19–30 years, there were highly significant positive associations for males and females aged 51–70 years, and all other subgroups had null or weak positive associations (*p* < 0.05) with no apparent pattern.

#### 3.3.3. Micronutrients of Public Health Concern and Micronutrients to Limit

Seventeen studies in 16 articles examined associations between dietary sugars and micronutrient intakes, with several articles reporting on multiple vitamins (for 14 articles, mean = 8 vitamins; range: 1 to 12) and multiple minerals (for 16 articles, mean = 5 minerals; range: 1 to 9). All 13 essential vitamins were examined, with vitamins A, B9, C, and D assessed most commonly (*n* = 12 articles each; see Table 3 and Appendix A). Eleven essential minerals were assessed, with the most common being calcium (*n* = 14), iron (*n* = 12), magnesium (*n* = 13), and zinc (*n* = 12). Seven articles reported on potassium and sodium, with four articles reporting on both. None of the included articles assessed associations between sugar intake and chloride, chromium, fluoride, molybdenum, or nickel. Most articles reported on added sugar associations with vitamins (*n* = 7) or minerals (*n* = 8), while total sugars and free sugars were reported less frequently (*n* = 4 each for vitamins; *n* = 5 each for minerals). Of the studies reporting on micronutrient intakes, most were in children or adolescent populations [16,17,18,19,26,30,35,38] or reported separate results for children and adults [14]. Six articles reported on studies in adult-only populations [13,22,23,31,33,36], while one was in a mixed population with children and adults [25]. While most studies showed only moderate potential for bias due to the study design (neutral ROB), studies identified as having high potential bias (poor ROB) measured dietary intake just once and failed to control for important confounders (see Appendix A).

##### Vitamin D

Thirteen studies from twelve articles reported on associations between dietary sugar and vitamin D intakes [14,16,17,18,19,22,25,26,30,31,35,36], and five showed high potential for bias [16,17,25,26,31]. Results reported for associations with added and free sugars were mostly negative, while associations with total sugars were mostly null (see Appendix A). Of eight studies reporting on added sugar, six controlled for total energy intake, and the results were mixed. Four of these studies from three articles reported negative associations between added sugars and vitamin D intakes in linear regression models (*p* ≤ 0.007 for all) [18,22,31]. One study with poor ROB showed null associations across quartiles of added sugar intake for both males and females, except for significantly lower vitamin D intakes in quartile 3 compared to quartile 1 of added sugar intake for females (*p* < 0.05) [25]. One study with poor ROB showed no correlation between vitamin D and added sugar intake overall but significantly higher vitamin D intake in quartiles 2, 3, and 4 of sugar intake compared to quartile 1 (*p* < 0.05) [26]. Three studies reported on free sugar associations with vitamin D intake [17,19,30]. Two of these were adjusted for total energy [17,19], and both were conducted in child and adolescent populations and showed negative associations between free sugars and vitamin D intakes (*p* < 0.01). All three studies assessing total sugar associations with vitamin D intake included results adjusted for total energy intake [16,30,36], and all three reported mostly null associations with one exception. One study reported null associations for eight age/sex subgroups but found a highly significant non-linear positive association for vitamin D (μg/1000 kcal) across quintiles of total sugars (%E) for adults overall after controlling for several covariates (*p* < 0.001) [36].

##### Calcium

Fifteen studies from 14 articles reported on associations between sugar and calcium intake [13,14,16,17,18,19,22,25,26,30,31,35,36,38], where five studies had high potential for bias [16,17,25,26,31]. Associations with added and free sugars were mostly negative, and associations with total sugars were positive. Of eight studies examining added sugar, six adjusted for energy in the design or analysis, and four of these (from three articles) reported negative associations with calcium in adjusted linear regression models [18,22,31]. One study with poor ROB reported null associations across quartiles of added sugar intake for both males and females except for a significantly lower calcium intake in quartile 4 compared to quartile 1 for females (*p* < 0.05) [25]. The last study, also with poor ROB, reported no relationship between added sugars and calcium based on Pearson’s partial correlations [26]. Four studies examined free sugars, with three of these adjusting for energy and finding mostly negative associations. One study found a negative trend in calcium intake across quintiles of free sugar intake in participants aged 4–10 years (*p* < 0.001) and 11–18 years (*p* < 0.01) [19], and another reported a negative trend across six free sugar cut-off levels (*p* < 0.001) [38]. One study with poor ROB found a non-linear association but significantly lower calcium intake in the highest compared to the lowest free sugar intake group (*p* < 0.05) [17]. Four other studies examined total sugar intake, with all four adjusting for energy and reporting generally positive associations with calcium intake [13,16,30,36]. One of these studies with poor ROB reported a null association for participants aged 9–13 years but positive associations for other age groups (2–8 years, *p* < 0.01; 14–18 years, *p* < 0.001) [16]. Another study reported a positive trend across quintiles of total sugar intake in the overall study population (*p* < 0.001) and across sex and age subgroups except for two null associations with calcium intake for males aged 19–30 years and 31–50 years [36].

##### Potassium

Eight studies from seven articles reported on potassium, and two studies had high potential for bias [16,17]. Three studies in two articles found negative associations with added sugars [14,22]. Two studies examining free sugar intake found non-linear associations with potassium intake but significantly lower potassium at the highest compared to the lowest free sugar intake levels [17,19]. One of these studies separately found a negative association with potassium intake in participants aged 11–18 years (*p* < 0.001) [19]. Three studies examined total sugars and found positive associations with potassium intake [13,16,36] except for two null associations in subgroup analyses for males aged 31–50 years and females aged 19–30 years [36].

##### Sodium

Seven studies reported on sodium, and two of these studies had high potential for bias [16,17]. Only one study examined added sugar intake and found no association with sodium measured by a dietary quality component score [33]. This same study similarly found no association between free sugar intake and the sodium component score. Two other studies found non-linear associations but significantly lower sodium intakes at the highest compared to the lowest free sugar intake levels [17,38]. Four studies that examined associations between total sugars and sodium intake found consistently negative associations across total study populations and multiple subgroups [13,16,23,36].

### 3.4. Strength of Evidence

Despite our comprehensive database search and our inclusion criteria, which considered many study designs, only cross-sectional studies met the eligibility criteria for this review, which stipulated that studies must report associations between quantified dietary sugar intake and measures of diet quality. With only cross-sectional evidence available in the current published literature, the strength of the evidence is very low for the association between dietary sugar intake and measures of diet quality, including dietary indexes and intakes of fiber, essential vitamins, and essential minerals.

## 4. Discussion

This systematic review included a comprehensive search of recent literature on associations between quantified intake of sugars from the daily diet and diet quality in generally healthy populations. Although multiple study designs were eligible, only cross-sectional studies met all inclusion criteria. Due to there being only cross-sectional evidence available in the current published literature, the strength of this body of evidence was rated very low for the association between intake of dietary sugars and measures of diet quality including dietary indexes and intakes of fiber, essential vitamins, and essential minerals. Most reviewed studies (~80%) found that higher intakes of added and free sugars were associated with lower diet quality, as measured by dietary indexes (mean Pearson’s *r* ranging from −0.13 to −0.42). This conclusion was found for both dietary indexes that included a sugar component and those that did not; although, the former showed stronger associations with intake of sugars, as expected. Most studies showed the same negative associations between added and free sugars and intakes of nutrients of public health concern, including fiber (~69% of studies), vitamin D (~54%), calcium (~47%), potassium (~71%), and sodium (~71%). While five of these studies reported some non-linear associations, comparisons between the highest and lowest intakes of sugars indicated mostly negative associations across nutrients assessed (8 out of the 11 reported non-linear associations).

That said, quantities of added and free sugars assessed in the included studies varied widely, and statistical approaches also showed a great deal of heterogeneity in that some studies examined thresholds of intake based on sugars as a percent of total energy consumed while others divided intakes into quantiles based on the study population range of intakes. While this review did not investigate specific intake levels (e.g., based on grams of sugars), a potential dose-response relationship between intake of added and free sugars and diet quality was indicated by four studies that compared different intake levels and reported increasingly negative associations with diet quality across higher intakes of sugars. Additionally, the pairwise comparisons reported in studies of nutrient intakes generally indicated that lower to moderate intakes of added and free sugars may have different magnitudes or directions of association with diet quality. These differences may be further complicated by age and sex, as many included studies showed different magnitudes or directions of association for male and female participants and for participants from different age groups. Despite this, findings from included studies that took place in children or adolescents were generally similar to those from studies in adults. However, the range of absolute or relative intake at which added and free sugars are associated with lower diet quality outcomes in different population subgroups has yet to be determined. Results for the intake of total sugars with diet quality measures were mixed, and the magnitude of associations between total sugars and dietary indexes was generally smaller than reported associations with added and free sugars. This is likely due to the fact that total sugars include naturally occurring sugars from foods that contribute important nutrients to the diet, such as whole fruits, vegetables, grains, and dairy.

Our findings are consistent with an older systematic review, which included studies published between 1972 and 2012 [4]. Similar to the prior systematic review, the most significant limitation of the present review is that the included studies were highly heterogeneous, which made meta-analysis unfeasible. While we were able to convert reported results to Pearson’s *r* correlations for the dietary index outcome, we did not perform meta-analysis to combine *r* values because such meta-analysis results would likely amplify the biases in the original studies. Other limitations of the present systematic review originated from measurement errors or biases in dietary assessments used by the included studies. Specifically, there is currently no standardized nutrient database or estimation method for quantifying intake of dietary sugars, and the included studies used a variety of databases and methods. Additionally, different diet quality indexes are based on different standards or recommendations and thus have different scoring compositions. Because of this, results from studies of different diet quality indices are not directly comparable.

Analyses of the relationship between intake of sugars and diet quality indexes, fiber, or micronutrient intakes are further complicated by measurement error or biases in dietary assessment tools, as well as the approaches used to account for the influence of total energy intake on the associations examined. Although we collected information on whether the original studies adjusted for total energy intake in analyses, we were not able to use meta-analytical techniques to quantitatively compare studies controlling for or not controlling for total energy intake. Moreover, counterarguments exist for whether energy adjustment is appropriate, especially when examining the relationship between added sugars and micronutrient intakes. These arguments indicate that controlling for total energy intake, particularly when micronutrient intake was analyzed using the nutrient density approach (%E), biases the results [39]. Because total energy is the denominator of nutrient density measures and total energy and energy from added sugar intake are strongly related, it is impossible to separate the association of added sugars with micronutrient consumption from that of total energy or energy from other sources. As for studies using dietary indexes, many of their algorithms already incorporate energy intake into scoring, so these studies may not need to separately adjust for energy in their analyses. For other indexes that do not incorporate energy into scoring, statistical analyses that utilize the residual method are recommended to remove correlations between higher energy intake and diet quality itself.

With these limitations in mind, the current body of evidence suggests that the intake of added or free sugars is a more reliable indicator of poor diet quality than the intake of total sugars, which include naturally occurring sugars in fruits, milk, and other foods containing additional nutrients to boost diet quality. However, the magnitude of associations between added or free sugars and diet quality measures is small to moderate, with concerns of confounding. Thus, the clinical significance of these findings is unclear. To strengthen the evidence base, a standardized country-specific database of added sugar values is urgently needed to allow between-study comparisons, at least for studies performed in the same country. Moreover, prospective studies, including trials, are needed to minimize confounding and investigate causal relationships to inform dietary guidelines and public health policies promoting nutrient-dense foods and a limited intake of added and free sugars.

## 5. Conclusions

This systematic review aimed to comprehensively examine the recent published literature on associations between quantified intake of dietary sugars and various measures of diet quality. Only cross-sectional studies were available and provided very low evidence suggesting that the highest intakes of added and free sugars may be associated with lower diet quality based on generally poorer dietary index scores and lower intakes of fiber, vitamin D, calcium, and potassium. Mixed results indicate that associations between total sugars and diet quality may be more nuanced and require some knowledge of specific food sources of sugars in the diet to determine a relationship. Current dietary guidelines recommend lower intakes of added and free sugars. Evidence from this review provides some support for such recommendations, but prospective studies will be necessary to establish a causal relationship between intake of sugars and diet quality.

## Figures and Tables

**Figure 1 nutrients-16-01549-f001:**
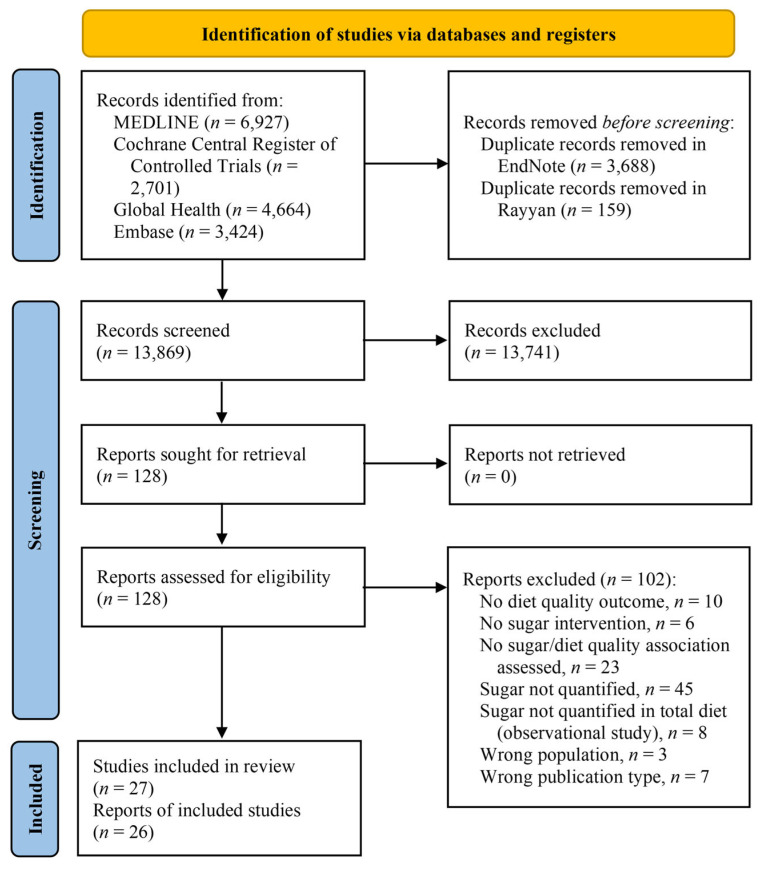
Flow diagram of the study search and selection process.

**Figure 2 nutrients-16-01549-f002:**
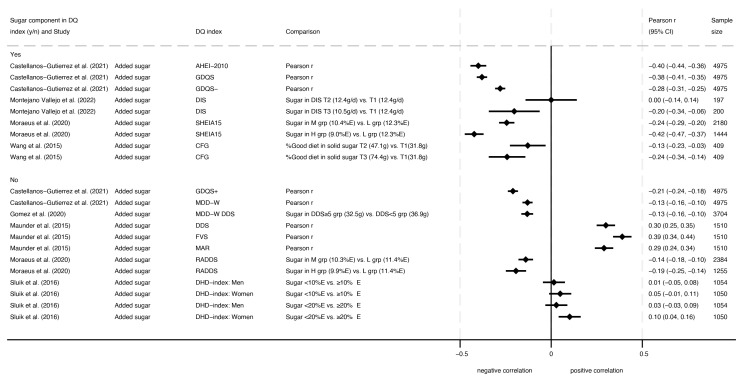
Forest plot with correlations for intake of added sugars and dietary indexes, where higher dietary index scores indicated higher diet quality [15,21,26,28,29,33,35].

**Figure 3 nutrients-16-01549-f003:**
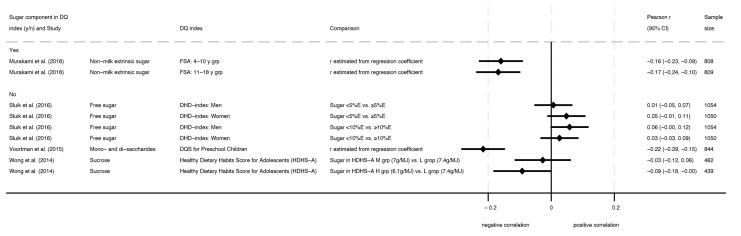
Forest plot with correlations for intake of free sugars and dietary indexes, where higher dietary index scores indicated higher diet quality [32,33,34,37].

**Figure 4 nutrients-16-01549-f004:**
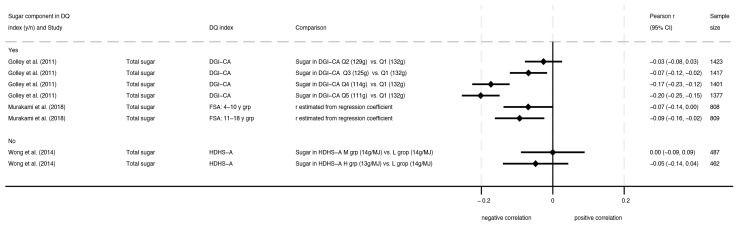
Forest plot with correlations for intake of total sugars and dietary indexes, where higher dietary index scores indicated higher diet quality [20,32,37].

**Table 1 nutrients-16-01549-t001:** Criteria used to determine eligibility for inclusion in this review.

Category	Inclusion Criteria	Exclusion Criteria
Study design	Intervention studies: Randomized controlled trials (parallel or crossover) Non-randomized controlled trials Quasi-experimental studies Before-and-after studies Cohort studies Nested case-control studies or case cohort studies Cross-sectional studies	Narrative review Systematic review Meta-analyses Letters to the editor Retrospective case-control studies Study protocols
Intervention or exposure	Any type of quantified dietary sugar from individuals’ total diets (not an exclusive list): monosaccharides disaccharides sugar-sweetened beverages corn syrup honey other unspecified dietary sugars	No quantifiable dietary sugar, including “sugar sweetened beverages (SSBs)” consumption where sugar amount was not quantified Sugar was quantified only in part of the diet (e.g., breakfast or snacks) Sugar was quantified for the whole population and not at the individual level Intravenous sugar administration
Outcomes	Measures of individual diet quality from the total diet: Micronutrient intake (essential vitamins and minerals) Fiber intake Diet quality indexes	No measures of diet quality Measures of diet quality only in part of the diet (e.g., breakfast, snacks) Diet quality was measured in the whole population but not at the individual level
Date of publication	January 2010 to October 2022	Studies published prior to 2010
Language of publication	Studies published in English	Articles published in languages other than English
Study participants	Generally healthy human participants	Animal subjects In-vitro and cell studies >20% of the study population has a disease (e.g., T2D, metabolic syndrome)
Age of study participants	>23 months old	≤23 months old

**Table 2 nutrients-16-01549-t002:** Study and participant characteristics of included studies.

Study, Year (Ref); Funding Type	Study Population Country; Data Source	Sample Size Analyzed/Enrolled (Male %)	Mean or Median Age (SD), y; Age Range, y	Dietary Assessment Tool(s); Use of Multiple Assessments	Source for Estimating Sugar Intake	Sugar Type(s) Examined (Unit)	Diet Quality Outcome(s) Examined
Castellanos-Gutierrez et al., 2021 [15]; non-profit	Mexico; Mexican National Health and Nutrition Survey (ENSANUT), 2012 and 2016	7517/15,227 (0%)	~29.8 (NR); 15–49	24HR (group 1), FFQ (group 2); none	National Institute of Public Health (INSP) food-composition table	Added sugars (g)	Indexes: Alternate Healthy Eating Index-2010 (AHEI-2010), Global Diet Quality Score (GDQS), Global Diet Quality Score positive submetric (GDQS+), Global Diet Quality Score negative submetric (GDQS−), Minimum Dietary Diversity for Women (MDD-W)
Chiavaroli et al., 2022 [16]; none	Canada; Canadian Community Health Survey (CCHS), 2004 and 2015	5491/5491 (50%)	~10 (NR); 2–18	24HR; none	2015 Canadian Nutrient File and Health Canada Bureau of Nutritional Science (BNS) food codes	Total sugars (%E)	Nutrients: fiber, vit A, thiamin, riboflavin, niacin, folate, vit B12, vit C, vit D, calcium, iron, magnesium, phosphorus, potassium, sodium, zinc
Fujiwara et al., 2020 [17]; government	Japan; National Health and Nutrition Survey Japan (NHNS), 2016	2919/4595 (51.6%)	9.7 (NR); 1–19	Weighed household dietary record (1 d); none	Comprehensive food composition database for common Japanese food items included in the Standard Tables of Food Composition in Japan (STFCJ)	Free sugars (%E)	Nutrients: fiber, vit A, thiamin, riboflavin, niacin, pantothenic acid, vit B6, folate, vit B12, vit C, vit D, vit E, vit K, calcium, copper, iron, magnesium, manganese, phosphorus, potassium, sodium, zinc
Fulgoni et al., 2019 [18]; industry	USA; National Health and Nutrition Examination Survey (NHANES), 2009–2014	7754/8583 (50.6%)	10 (NR); 2–18	24HR; two assessments averaged	USDA Food Patterns Equivalent Database (FPED) for each NHANES release	Added sugars (%E)	Nutrients: vit A, thiamin, riboflavin, niacin, vit B6, folate, vit B12, vit C, vit D, vit E, calcium, copper, iron, magnesium, phosphorus, selenium, zinc
Fulgoni et al., 2020 [14]; industry	USA; National Health and Nutrition Examination Survey (NHANES), 2011–2014	13,005/15,829 (NR)	NR (NR); 2+	24HR; two assessments averaged	USDA Food Patterns Equivalent Database (FPED) for each NHANES release	Added sugars (%E)	Nutrients: fiber, vit D, calcium, potassium
Gibson et al., 2016 [19]; industry	UK; UK National Diet and Nutrition Survey (NDNS), 2008–2012	2073/4156 (51.4%)	~9.75 (NR); 1.5–18	Dietary records; 3 to 4 d records averaged	In-house dietary assessment system of the Medical Research Council Human Nutrition Research, DINO (Data In, Nutrients Out)	Free sugars (%E) ^1^	Nutrients: non-starch polysaccharides (NSP) fiber, vit A, thiamin, riboflavin, niacin equivalents, vit B6, folate, vit B12, vit C, vit D, vit E, calcium, copper, iodine, iron, magnesium, potassium, selenium, zinc
Goletzke et al., 2015 [13]; government, non-profit	Australia; Pregnancy and Glycemic Index Outcomes (PREGGIO) study members, February 2010 to September 2012	566/691 (0%)	29.8 (IQR: 26.6, 33.3); NR	Food records; 3 d records averaged at baseline and at 34 wk of gestation	Customized database incorporating Australian food composition tables and published Glycemic Index values (FoodWorks 2009 Professional edition, version 6.0.2539; Xyris Software)	Total sugars (g/1000 kJ)	Nutrients: fiber, vit A equivalents, thiamin, riboflavin, niacin equivalents, total folate, vit C, calcium, iron, magnesium, potassium, sodium, zinc
Golley et al., 2011 [20]; government	Australia; Australian National Children’s Nutrition and Physical Activity Survey, 2007	3416/3601 (NR)	~10 (NR); 4–16	24HR; two assessments averaged	Nutrient composition database developed specifically for the survey (AUSNUT2007)	Total sugars (g)	Index: Dietary Guideline Index for Children and Adolescents (DGI-CA)
Gomez et al., 2020 [21]; industry, non-profit	Argentina, Brazil, Colombia, Costa Rica, Chile, Ecuador, Peru, Venezuela; Latin American Study on Nutrition and Health/Estudio Latino Americano de Nutrición y Salud (ELANS), September 2014–August 2015	3704/9218 (0%)	31.2 (NR); 15–49	24HR; two assessments averaged	Nutrition Data System for Research (NDS-R) software version 2014	Added sugars (g)	Index: Minimum Dietary Diversity for Women (MDD-W) Dietary Diversity Score (DDS)
González-Padilla et al., 2020 [22]; government, non-profit	Sweden; National Swedish Food Survey of Adults (Riksmaten Adults), May 2010 to July 2011	1797/2268 (44%)	48 (16.6); 18–80	Food diary; 4 d diaries averaged	The national food composition database	Added sugars (%E)	Nutrients: fiber, folate, vit C, vit D, calcium, iron, magnesium, potassium, selenium, zinc
	Sweden; Malmö Diet and Cancer Study (MDCS), March 1991 to October 1996	12,238/28,098 (45%)	57.6 (6); 45–68	Food diary, FFQ, and interview; 7 d diaries averaged	Malmö Food and Nutrient Database based on the Swedish Food Database PC KOST-93	Added sugars (%E)	Nutrients: fiber, folate, vit C, vit D, calcium, iron, magnesium, potassium, selenium, zinc
Gress et al., 2020 [23]; NR	USA; National Health and Nutrition Examination Survey (NHANES), 2001–2016	38,722/82,097 (0.5%)	43.6 (15.6); 18–75	24HR; two assessments averaged	USDA Food and Nutrient Database for Dietary Studies (FNDDS)	Total sugars (g)	Nutrient: sodium
Kaartinen et al., 2017 [24]; government	Finland; The DIetary, Lifestyle and Genetic determinants of Obesity and Metabolic syndrome (DILGOM) study, April to June 2007	4842/5024 (46.3%)	~52.6 (NR); 25–74	FFQ; none	In-house software and the Finnish National Food Composition Database (Fineli^®^)	Added sucrose and fructose (g), naturally occurring sucrose and fructose (g)	Nutrient: fiber
MacIntyre et al., 2012 [25]; NR	South Africa; Transition and Health during Urbanisation in South Africa (THUSA) survey, 1996–1998	1045/1742 (~45.2%)	~36.4 (NR); 15+	FFQ; none	The FoodFinder^®^ dietary analysis programme of the Medical Research Council of South Africa	Added sugars (%E, g)	Nutrients: fiber, vit A, thiamin, riboflavin, niacin, pantothenic acid, vit B6, biotin, folate, vit B12, ascorbic acid, vit D, vit E, calcium, iron, magnesium, phosphorus, zinc
Maunder et al., 2015 [26]; government, non-profit	South Africa; National Food Consumption Survey (NFCS), 1999	2818 (2200 weighted)/2894 (NR)	~4.95 (NR); 1–8.9	24HR; none	National food composition database	Added sugars (%E, g)	Indexes: Dietary Diversity Score (DDS), Food Variety Score (FVS), Mean Adequacy Ratio (MAR) ^2^ Nutrients: fiber, vit A, thiamin, riboflavin, niacin, pantothenic acid, vit B6, biotin, folic acid, vit B12, vit C, vit D, vit E, calcium, iron, magnesium, phosphorus, zinc
Meira et al., 2021 [27]; government	Brazil; Campinas Health Survey, 2014–2015, and the Campinas Food Intake and Nutritional Status Survey (Campinas Nutrition Survey), 2014–2015	914/1023 (52%)	14.6 (CI95%: 14.4 to 14.8); 10–19	24HR; none	Nutrition Data System for Research (NDS-R, version 2015, Nutrition Coordinating Center, University of Minnesota)	Free sugars (%E)	Nutrient: fiber
Montejano Vallejo et al., 2022 [28]; government	Germany; DONALD (Dortmund Nutritional and Anthropometric Longitudinal Designed) Study, 1985 to January 2021	298/1761 (52%)	15.1 (NR); 15–17.1	Weighed dietary records; 2 to 5 × 3 d records averaged	LEBTAB nutrient database (based on the German standard food tables)	Added sugars (%E)	Index: Dietary Index (DI) score (internally developed based on the EAT–Lancet Reference Diet)
Moraeus et al., 2020 [29]; none	Sweden; Riksmaten Adolescents, 2016–2017	2905/3477 (~43.7%)	~14.5 (NR); NR	24HR; two assessments averaged	Swedish Food Agency (SFA) food composition database, version Riksmaten adolescents 2016–2017	Added sugars (%E)	Indexes: Swedish Healthy Eating Index for Adolescents 2015 (SHEIA15), Riksmaten Adolescents Diet Diversity Score (RADDS)
Morales-Suarez-Varela et al., 2020 [30]; none	Spain; Anthropometry and Child Nutrition of Valencia (ANIVA) Study, 2013–2014	2237/2563 (49.1%)	7.3 (NR); 6–8	Food journal; 3 d journal (unclear if averaged)	DIAL software for diet assessment and food calculations (Department of Nutrition (UCM) & Alce Ingeniería, S.L. Madrid, Madrid, Spain)	Simple sugars (g), total sugars (%E)	Nutrients: fiber, vit A, vit B1 (thiamine), vit B2 (riboflavin), vit B6, folate, vit B12, vit C, vit D, vit E, calcium, iodine, iron, magnesium, zinc
Moshtaghian et al., 2016 [31]; government	Australia; Blue Mountains Eye Study 4 (BMES4), 2007–2009	879/1149 (42.9%)	~76.2 (NR); 49+	FFQ; none	NUTTAB2010 Australian food composition database	Added sugars (%E)	Nutrients: fiber, vit A (retinol equivalents), thiamin, riboflavin, vit B6, dietary folate equivalents, vit B12, vit C, vit D, vit E, calcium, iodine, iron, magnesium, zinc
Murakami et al., 2018 [32]; government	UK; National Diet and Nutrition Survey (NDNS), January 1997 to December 1997	1617/2127 (~50.9%)	~10.65 (NR); 4–18	Weighed dietary records; 7 d records averaged	British Food Standards Agency (FSA) nutrient databank	Total sugars (%E), non-milk extrinsic sugars (%E)	Index: British Food Standards Agency (FSA) nutrient profiling system score
Sluik et al., 2016 [33]; government	The Netherlands; Dutch National Food Consumption Survey (DNFCS), 2007–2010	3817 total (2104 adults)/3819 (50%)	~38 (NR); 7–69	24HR; two assessments averaged	Dutch National Food Composition Table 2011	Added sugars (%E), free sugars (%E)	Index: Dutch Healthy Diet-index (DHD-index) Nutrients: fiber, sodium
Voortman et al., 2015 [34]; government, non-profit	The Netherlands; Generation R Study, 2001–2005	844/844 (~49%)	2.1 (95% CI: 2–2.33); NR	FFQ; none	Dutch Food Composition Table 2006	Monosaccharides and disaccharides (g)	Index: Diet Quality Score for Preschool Children (diet score)
Wang et al., 2015 [35]; government, non-profit	Canada; QUebec Adipose and Lifestyle InvesTigation in Youth (QUALITY) study, 2005–2008	613/630 (54.4%)	9.6 (0.9); 8–10	24HR; three assessments averaged	USDA Database for the Added Sugars Content of Selected Foods (USDA 2006)	Added sugars (g)	Indexes: Canadian Healthy Eating Index (HEI-C), Canada’s Food Guide for “Good diet” Nutrients: vit A, vit D, calcium, magnesium, phosphorus
Wang et al., 2020 [36]; none	Canada; Canadian Community Health Survey (CCHS)-Nutrition, 2015	11,817/20,487 (50.6%)	49.4 (SE = 0.3); 19+	24HR; none	2015 Canadian Nutrient File and Health Canada Bureau of Nutritional Science (BNS) food codes	Total sugars (%E)	Nutrients: fiber, vit A, thiamin, riboflavin, niacin, vit B6, folate, vit B12, vit C, vit D, calcium, iron, magnesium, phosphorus, sodium, zinc
Wong et al., 2014 [37]; government	New Zealand; New Zealand (NZ) Adult Nutrition Survey, 2008–2009	694/695 (46.8%)	16.5 (SE = 0.1); 15–18	Dietary habits questionnaire and 24 HR; none	NZ Food Composition Database or overseas food composition data when appropriate for nutrient estimation	Total sugars, sucrose, fructose, glucose, lactose, maltose (g/MJ for each)	Index: Healthy Dietary Habits Score for Adolescents (HDHS-A)
Wong et al., 2019 [38]; none	Australia; Australian Health Survey, 2011–2012	1527 (1466 weighted)/2812 (51.2%)	9.7 (4.9); 2–18	24HR; two assessments averaged	Australian Food and Nutrient (AUSNUT) 2011–2013 food composition database	Free sugars (%E)	Nutrients: fiber, vit A (as retinol equivalents), thiamin, riboflavin, folate (as dietary folate equivalent), vit C, vit E, calcium, iodine, iron, magnesium, phosphorus, potassium, sodium, zinc

24 HR, 24 h recall; %E, percent of total energy intake; d, day(s); FFQ, food frequency questionnaire; g, grams; IQR, inner quartile range; NR, not reported; SE, standard error; USDA, U.S. Department of Agriculture; vit(s), vitamin(s); wk, week(s). ^1^ Reported using non-milk extrinsic sugars as a proxy for free sugars. ^2^ The Mean Adequacy Ratio (MAR) was calculated for 11 vitamins (A, B6, B12, C, thiamin, riboflavin, niacin, and folate) and minerals (calcium, iron, and zinc).

**Table 3 nutrients-16-01549-t003:** Summary of sugar types and dietary outcome associations reported by 26 included articles.

Dietary Outcome	Total Articles, *n* (%)	Added Sugars, *n*	Free Sugars, *n*	Total Sugars, *n*
Indexes	11 (42.3)	7	4	3
Fiber ^1^	15 (57.7)	7	7	4
Essential vitamins	14 (53.8)	7	4	4
A	12 (46.2)	5	4	4
B1 (thiamin)	11 (42.3)	4	4	4
B2 (riboflavin)	11 (42.3)	4	4	4
B3 (niacin)	8 (30.8)	3	2	3
B5 (pantothenic acid)	3 (11.5)	2	1	0
B6 (pyridoxine)	8 (30.8)	4	3	2
B7 (biotin)	2 (7.7)	2	0	0
B9 (folate)	12 (46.2)	5	4	4
B12	9 (34.6)	4	3	3
C	12 (46.2)	5	4	4
D ^1^	12 (46.2)	7	3	3
E	8 (30.8)	4	4	1
K	1 (3.8)	0	1	0
Essential minerals	16 (61.5)	8	5	5
Calcium ^1^	14 (53.8)	7	4	4
Chloride	0 (0.0)	0	0	0
Chromium	0 (0.0)	0	0	0
Copper	3 (11.5)	1	2	0
Fluoride	0 (0.0)	0	0	0
Iodine	4 (15.4)	1	3	1
Iron	12 (46.2)	5	4	4
Magnesium	13 (50.0)	6	4	4
Manganese	1 (3.8)	0	1	0
Molybdenum	0 (0.0)	0	0	0
Nickel	0 (0.0)	0	0	0
Phosphorus	8 (30.8)	4	2	2
Potassium ^1^	7 (26.9)	2	3	2
Selenium	3 (11.5)	2	1	0
Sodium ^2^	7 (26.9)	1	3	4
Zinc	12 (46.2)	5	4	4

^1^ Dietary components of public health concern for the general U.S. population [10]. ^2^ Dietary component to limit for the general U.S. population [10].

## Data Availability

Data supporting reported results can be found in the Appendix A.

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
