# Peer review of "Associations between Intake of Dietary Sugars and Diet Quality: A Systematic Review of Recent Literature"

_nutrients, 2024, doi:10.3390/nu16111549_

Round 1

Reviewer 1 Report

Comments and Suggestions for Authors

The authors conducted a systematic review that synthesized recent literature on associations between sugar intake and diet quality in generally healthy populations aged 2 years or older.
They followed the required steps to perform a systematic review, and the assembled data that was carefully synthesized is very noteworthy.
The only issue I have with the work is the grammar. They are too many places that need improvement, which would directly elevate the scientific soundness of the work.
Please authors, try to meticulously correct the grammar, avoid repetitions, and utilize a professional English speaker as a guide. Look forward to your revised manuscript.

Comments on the Quality of English Language

Please, utilize a professional English speaker to revise the work.

Author Response

Thank you for these comments. We have thoroughly checked the grammar throughout the document and made changes to improve clarity. As for repetitious language, we have reviewed the entire text and believe any remaining repetitions serve to clearly explain the results and assist readers. If additional concerns remain, please provide specific examples of where grammar or repetitious language is an issue, as our professional English speakers identified very few grammatical errors.

Reviewer 2 Report

Comments and Suggestions for Authors

Main research question:

The main topic of the study is the association between sugar intake (added, free, or total) and diet quality in usually healthy populations. It uses a thorough evaluation of recent literature to determine how different forms of sugar consumption connect with dietary indices or micronutrient intakes.

Originality and relevance:

The topic is quite timely, particularly considering the growing public health concern for diet-related chronic disorders. It fills a specific need by updating and critically evaluating current evidence since the last major assessments, which are now regarded as outdated. This considerably advances the field of medicine, particularly given the systematic use of previously unavailable risk-of-bias evaluations.

Contribution to subject area:

The paper presents a more recent and complete synthesis of research on the relationship between sugar consumption and diet quality. It improves by employing a rigorous systematic approach, incorporating a greater range of study types, and conducting a more recent literature search.

Methodological improvements:

The authors should try widening the types of sugars studied; differentiating within more specific types of sugars and their sources (for example, sugars in fruits vs. added sugars in processed foods) might give more nuanced insights into how different forms of sugar consumption impact diet quality.

Practical implementation recommendations: Including information on how the findings might be applied to public health policies, dietary guidelines, and medical treatments in the Discussions section would make the study more relevant to policymakers and practitioners.

Consistency of conclusions:

The conclusions are conservatively worded and consistent with the evidence given, recognising the limitations inherent in nearly all of the study designs included in the evaluation. The manuscript rightly calls for a Conclusion section to enhance clarity for the readers.

Appropriateness of references:

The citations are adequate and appear to provide comprehensive coverage of the relevant background and previous research.

Comments on tables and figures:

The tables and figures are well-designed and present clear, detailed data synthesis and analysis.

Author Response

Methodological improvements:

The authors should try widening the types of sugars studied; differentiating within more specific types of sugars and their sources (for example, sugars in fruits vs. added sugars in processed foods) might give more nuanced insights into how different forms of sugar consumption impact diet quality.

Response: Thank you for this suggestion, and we agree this would be helpful. Unfortunately, the studies we reviewed typically did not differentiate sugar sources at such a granular level. To our recollection, only one study in our review reported specific sugars which could theoretically be linked to more specific dietary sources (e.g., fructose, lactose), but even in this study, the exact sources of sugars were not identified and could vary widely by sugar type (e.g., fructose in whole fruits versus fruit juices or snacks sweetened with fruit concentrates). All other studies analyzed sugars that were more broadly combined across dietary sources. Therefore, we believe our review provided the most granular results currently available by separating our analyses by the three commonly reported sugar types (added, free, and total) and by separately analyzing indexes with and without a sugar component.

Practical implementation recommendations: Including information on how the findings might be applied to public health policies, dietary guidelines, and medical treatments in the Discussions section would make the study more relevant to policymakers and practitioners.

Response: Thank you for this comment. Due to the very low evidence and high heterogeneity currently present in the published literature, we do not believe the evidence is suitable to support policies, guidelines, or treatments at this time. This position is stated in our final sentence of the discussion which says, “Moreover, prospective studies, including trials, are needed to minimize confounding and investigate causal relationships to inform dietary guidelines and public health policies promoting nutrient-dense foods and limited intake of added and free sugars.”

Consistency of conclusions:

The conclusions are conservatively worded and consistent with the evidence given, recognizing the limitations inherent in nearly all of the study designs included in the evaluation. The manuscript rightly calls for a Conclusion section to enhance clarity for the readers.

Response: Thank you for this suggestion. We added a conclusion paragraph.